# Green Veterinary Pharmacology for Honey Bee Welfare and Health: *Origanum heracleoticum* L. (Lamiaceae) Essential Oil for the Control of the *Apis mellifera* Varroatosis

**DOI:** 10.3390/vetsci9030124

**Published:** 2022-03-09

**Authors:** Fabio Castagna, Roberto Bava, Cristian Piras, Cristina Carresi, Vincenzo Musolino, Carmine Lupia, Mariangela Marrelli, Filomena Conforti, Ernesto Palma, Domenico Britti, Vincenzo Musella

**Affiliations:** 1Department of Health Sciences, University of Catanzaro Magna Græcia, CISVetSUA, 88100 Catanzaro, Italy; fabiocastagna@unicz.it (F.C.); roberto.bava@unicz.it (R.B.); c.piras@unicz.it (C.P.); studiolupiacarmine@libero.it (C.L.); palma@unicz.it (E.P.); britti@unicz.it (D.B.); musella@unicz.it (V.M.); 2Interdepartmental Center Veterinary Service for Human and Animal Health, University of Catanzaro Magna Græcia, CISVetSUA, 88100 Catanzaro, Italy; 3Department of Health Sciences, Institute of Research for Food Safety & Health (IRC-FISH), University of Catanzaro Magna Græcia, 88100 Catanzaro, Italy; 4Pharmaceutical Biology Laboratory, Department of Health Sciences, Institute of Research for Food Safety & Health (IRC-FISH), University of Catanzaro Magna Græcia, 88100 Catanzaro, Italy; 5National Etnobotanical Conservatory, Castelluccio Superiore, 85040 Potenza, Italy; 6Mediterranean Etnobotanical Conservatory, Sersale (CZ), 88054 Catanzaro, Italy; 7Department of Pharmacy, Health and Nutritional Sciences, University of Calabria, Rende, 87036 Cosenza, Italy; mariangela.marrelli@unical.it (M.M.); filomena.conforti@unical.it (F.C.); 8Nutramed S.c.a.r.l., Complesso Ninì Barbieri, Roccelletta di Borgia, 88021 Catanzaro, Italy

**Keywords:** honey bee welfare and health, *Apis mellifera*, varroatosis, *Varroa destructor*, green veterinary pharmacology, *Origanum heracleoticum* essential oil, acute toxicity, fumigant toxicity, repellent action

## Abstract

Varroatosis, caused by the *Varroa destructor* mite, is currently the most dangerous parasitic disease threatening the survival of honey bees worldwide. Its adverse effect on the welfare and health of honey bees requires the regular use of specific acaricides. This condition has led to a growing development of resistance phenomena towards the most frequently used drugs. In addition, another important aspect that should not be understated, is the toxicity and persistence of chemicals in the environment. Therefore, the identification of viable and environmentally friendly alternatives is urgently needed. In this scenario, essential oils are promising candidates. The aim of this study was to assess the contact toxicity, the fumigation efficacy and the repellent effect of *Origanum heracleoticum* L. essential oil (EO) against *V. destructor* mite. In the contact tests, each experimental replicate consisted of 15 viable adult female mites divided as follows: 5 treated with EO diluted in HPLC grade acetone, 5 treated with acetone alone (as negative control) and 5 treated with Amitraz diluted in acetone (as positive control). The EO was tested at concentrations of 0.125, 0.25, 0.5, 1 and 2 mg/mL. For each experimental replicate, mortality was manually assessed after one hour. The efficacy of EO fumigation was evaluated through prolonged exposure at different time intervals. After each exposure, the 5 mites constituting an experimental replicate were transferred to a Petri dish containing a honey bee larva and mortality was assessed after 48 h. The repellent action was investigated by implementing a directional choice test in a mandatory route. During the repellency tests the behavior of the mite (90 min after its introduction in the mandatory route) was not influenced by the EO. In contact tests, EO showed the best efficacy at 2 and 1 mg/mL concentrations, neutralizing (dead + inactivated) 90.9% and 80% of the mites, respectively. In fumigation tests, the mean mortality rate of *V. destructor* at maximum exposure time (90 min) was 60% and 84% at 24 and 48 h, respectively. Overall, these results demonstrate a significant efficacy of *O. heracleoticum* EO against *V. destructor*, suggesting a possible alternative use in the control of varroatosis in honey bee farms in order to improve *Apis mellifera* welfare and health and, consequently, the hive productions.

## 1. Introduction

To date, varroatosis is the main parasitic disease of honey bees, with a great impact on the welfare and health of bees and consequently, on their productivity and performance. This disease is caused by the *Varroa destructor* mite, which was assumed to be *Varroa jacobsoni* before the last century [1,2]. Since its first infestation against *Apis mellifera*, the mite has spread rapidly all over the world, profoundly changing the approach to beekeeping [2]. This ectoparasite, in addition to exerting an activity of depletion of essential nutrients for the proper physiological maintenance of the honey bee, is vector of many viruses during its meal [1].

The risk associated to a reduced survival of honey bee colonies is mainly related to the viral vector action of the parasite. This peculiarity means that *V. destructor* is often indicated as the main cause of the collapse of the hives [3,4,5].

Following a high-grade infestation, the transmission of viruses, such as acute paralysis virus (ABPV) and deformed wing virus (DWV), increases [6,7,8]. To prevent virus outbreaks, control strategies against mite populations are essential for beekeepers [6]. Pharmaceutical preparations currently on the market are formulated using synthetic acaricides, organic acids and essential oils (i.e., ApiLifeVar^®^, Chemicals Laif, SpA, Vigonza, PD, Italy), prepared with thymol, eucalyptus and menthol essential oils). Oxalic acid and Amitraz are the most widely used drugs among organic acids and synthetic acaricides, respectively [9]. The growing trend in the use of synthetic acaricides has been dictated by their ease of use and a formulation that allows them to cover two brood cycles.

However, medicinal products containing such active ingredients are not free from side effects. Very low doses/concentrations have been demonstrated to have a sublethal effect on the physiology, neurology, metabolism and/or behavior of honey bees [10]. Furthermore, chemicals can remain in beehive products and this accumulation can lead to chronic exposure of both adult honey bees and their immature forms to sublethal doses of acaricides [11,12]. This can be detrimental to the fate of the colony because the sub-lethal effects can result in progressive depopulation of the hive [13,14].

Pettis et al. (2013) showed that consumption of pollen contaminated with fungicides (chlorothalonil or pyraclostrobin) and acaricides (2,4 dimethylphenyl formamide, a metabolite of Amitraz, bifenthrin or fluvalinate) can double the relative risk of *Nosema* infection [15]. This observation is supported by the evidence gathered by Cutler (2013) [16], who found that worker honey bees exposed to Amitraz show a higher defecation rate than untreated honey bees. Therefore, there is a strong correlation between N-2,4-dimethylphenyl-N-methylformamidine residues (intermediate metabolites of Amitraz) and *Nosema* levels in honey bee samples [16]. Furthermore, low doses of Amitraz double the heart rate of honey bees and impair their reactivity to the noxæ pathogenic virus [17]. Drones exposed to cumaphos or fluvalinate have also been shown to have reduced sperm counts and viability and suffer from reduced body weight [18].

On the other hand, organic acids do not remain in the hive products but are toxic to adult honey bees and larvae. Following contact with oxalic acid, honey bees show a shorter life span and less active behavior within the colony [19,20].

The identification of alternative methods for the control of *V. destructor* becomes urgent, also in the light of the increasingly and widespread phenomena of drug resistance reported [21,22,23,24,25]. Control methods based on the use of essential oils offer considerable promise due to their low toxicity to humans and the environment [26].

Essential oils are low molecular weight volatile substances produced by the secondary metabolism of plants. They are generally composed of a complex mixture of mono- and sesquiterpene hydrocarbons, oxygenated materials, phenyl proponoids and other compounds [27].

Thanks to their particular and diversified chemical composition, the parasitic populations treated with essential oils are often unable to develop drug resistance phenomena [26,28,29].

Many essential oils have produced excellent results in controlling the growth of many species of insects and parasites that are harmful to crops, food and animals [30,31].

In this perspective, the ethnobotanical knowledge are extremely useful in choosing the most promising plant extracts [32]. Thyme essential oil has provided the most promising results against *V. destructor* mite and is the most represented active ingredient in products on the market [33].

Other essential oils have also been found to be effective in both in vitro and in vivo tests [34] such as that derived from the perennial herb plant of oregano. It looks similar to a small shrub of variable height depending on the species. Oregano (*Origanum* spp.) grows naturally in sunny arid regions but is also grown as an aromatic plant and for its therapeutic properties. It is an aromatic plant native to western and southwestern Eurasia and the Mediterranean subregion [35]. Known worldwide as a fresh or dried spice in the culinary field, it is also used as an antioxidant additive or preservative in many foods [36].

There are several species belonging to the genus *Origanum* and, among these, the best known are *O. vulgare* and *O. majorana* [37,38]. The oregano plant species that grow in Italy are: *O. vulgare*, *O. heracleoticum*, *O. majorana* and *O. onites* [39]. The common oregano species *O. vulgare* can be found in central and northern Italy, while it is lacking in southern Italy. On the contrary, the species *O. heracleoticum* grows spontaneously in southern Italy [40,41].

The essential oils extracted from the plant show antimicrobial and antifungal activity against human, plant and food-borne pathogens [42,43]. It has been tested for the control of different species of beetles, diptera and lepidoptera with promising results [44].

In general, the essential oils isolated from plants of the genera *Oreganum* have proved to be excellent alternative tools for controlling the *V. destructor* mite. In one of the most recent studies conducted by Sabahi et al. (2017), a particular method of administering essential oils to honey bee colonies was evaluated. Electric vaporizers containing oregano essential oil (Sigma^®^, Missasauga, ON, Canada), were installed above the brood chamber and, running continuously, resulted in a 97–98% reduction in the mite population of the hive [45].

Based on these results, the aim of the present study was to investigate the in vitro activity of the essential oil of oregano, referring to the less studied species *Origanum heracleoticum*., against the ectoparasitic mite *V. destructor*. The data obtained could give further boost to the commercialization and use of a pharmacological product based on essential oil of oregano against *V. destructor* infestations.

## 2. Materials and Methods

The experiments were carried out in the parasitology laboratory of the Interdepartmental Center Veterinary Service for Human and Animal Health (CISVetSUA), University “Magna Graecia” of Catanzaro, in the month of July 2021.

Two apiaries in the province of Catanzaro (Calabria Region in southern Italy), heavily infested naturally with *V. destructor*, were used as a source of mites. Colonies enrolled in the study were not treated with acaricides in the previous six months.

In brief, several frames in which the drone brood had been reared, were transported from the apiary to the mite harvesting laboratory.

The collection of the mite was performed in the laboratory as described below. Each brood cell in the frame was deprived of the wax operculum that ensured its closure and was inspected. If mites were present, they were picked up with a fine paintbrush and moved into Petri dishes with live honey bee larvae and pupae to prevent starvation during harvesting operations. This method is time-consuming, but allows for the collection of less traumatized mites than those would be obtained with other methods, such as the sugar smoothie method [46].

The tests were performed immediately after the collection of the mites. Prior to each test, mites that appeared to be newly molted, weak or abnormal were excluded because they may have had a different response in bioassays.

### 2.1. Plant Material and Extraction Technique

The aerial parts of *O. heracleoticum* were harvested in June directly in the field in a natural growth area on the Ionian side of the Pollino massif (southern Italy), at an average altitude of 450 m above sea level. This autochthonous *Origanum* species typically grows on the eastern spurs of the Pollino National Park, on the Calabrian side.

The taxonomic identification was confirmed by Dr. V. Musolino and Dr. C. Lupia, Department of Health Sciences, University “Magna Graecia” of Catanzaro. A specimen voucher is deposited at the Ethnobotanical Conservatory of Castelluccio Superiore, Potenza, Italy, under position number 48 of the Labiatae family.

To obtain the essential oil, the fresh plant material was washed and extracted by steam distillation for 2 h, using a Clevenger-type apparatus (Albrigi Luigi, Verona, Italy). The essential oil obtained was dried over anhydrous sodium sulfate and stored at +4 °C until needed.

### 2.2. Gas Chromatography-Mass Spectrometry (GC-MS) Analyses

The chemical composition of the essential oil was assessed by gas chromatography-mass spectrometry (GC-MS). Analyses were performed using a Hewlett-Packard 6890 gas chromatograph (Agilent, Milan, Italy) equipped with an SE-30 capillary column (100% dimethylpolysiloxane; 30 m length; 0.25 mm internal diameter; 0.25 μm film thickness) coupled to a Hewlett Packard 5973 mass spectrometer (Agilent, Milan, Italy). Analyses were performed with helium as carrier gas (linear velocity, 0.00167 cm/s) using a programmed temperature (from 60 to 280 °C, at a rate of 16 °C/min). The injector and detector were set to 250 °C and 280 °C, respectively [47]. The mass spectra of the detected molecules were compared with the Wiley 138 mass spectra library to identify the constituents of the essential oils.

### 2.3. Acute Toxicity towards the Mites

The methods of Gashout and Guzmán-Novoa (2009) [48], adapted from Bava et al. (2021) [26], to assess the acute toxicity of essential oil for mites have been used.

For each daily test, at least 100 mites were collected to establish experimental replicates. The frames were removed from the original colonies and transported to the laboratory. Previously, each cell of the frames was inspected for mites. The mites found were collected with a fine paintbrush and transferred into a Petri dish with live honey bee larvae.

The tested essential oil and the active ingredient Amitraz (Merck, 45323) were diluted in HPLC grade acetone to a concentration of 2 mg/mL, 1 mg/mL, 0.5 mg/mL, 0.25 mg/mL and 0.125 mg/mL. Amitraz and acetone alone were used as positive and negative controls, respectively.

The eppendorf tubes (2 mL) were filled with 50 µL of essential oil diluted in acetone and placed open in the oven to evaporate the diluent. The tubes were often rolled up on the walls to allow evaporation of acetone and to coat the walls of the tubes with the essential oil. This process was repeated for 15–20 min. As verified by Gashout and Guzmán-Novoa (2009) [48], it is unlikely that a significant amount of the tested product has evaporated within this time period due to its high boiling point, which exceeds 200 °C [49]. Subsequently, for each technical replicate and positive and negative control, five live adult female mites were gently transferred into the previously prepared tube using a fine paintbrush. Once the mites were transferred inside, the tubes were sealed and placed in a dark room at 34 °C and 65% of relative humidity. The temperature and humidity conditions set for incubation are the natural ones present at the brood level. In previous studies, these conditions were more conducive to the development and reproduction of the *Varroa* mite [26,50,51].

Ten technical replicates were set up for each concentration of tested solution, using for each of them Amitraz for the positive control and acetone for the negative control.

Acute toxicity was determined by recording mite mortality after 1 h of exposure. At the end of the hour, the parasites contained within each tube were transferred to a Petri dish and examined under a stereoscopic microscope. The mites were considered dead if they did not move when pushed. Mites that only moved one or more legs were classified as inactive [26]. The inactivity condition was considered equivalent to death. Dead and inactive mites were classified as neutralized.

### 2.4. Fumigant Toxicity towards the Mites

To verify the toxicity of the fumigation of the essential oil, a cotton swab was inserted into the recess on the inner surface of the cap which allows the hermetic closure of the Eppendorf tube. For each experimental replicate, 5 adult female mites were transferred to the bottom of a 2 mL Eppendorf tube using a fine paintbrush. A piece of tulle was immediately inserted into the test tube and interposed between the mites and the cap. This arrangement made it possible to avoid contact between the mites and the cotton present in the cap of the test tube. The cotton ball was then soaked with 40 µL of essential oil diluted in distilled water to a concentration of 1 mg/mL. The tube cap was hermetically sealed.

The Eppendorf tubes thus prepared were placed in an incubator at a temperature of 34 °C and 65% of relative humidity. The mites were exposed to the vapors for several times: 15, 30, 45 and 90 min [46].

In particular, the fumigation toxicity tests were conducted by exposing 5 mites to the vapors produced by 40 µL of 1 mg/mL essential oil (water dilution) into a final volume of 2 mL (final concentration 20 mg per liter of air). For the negative controls, the mites were transferred to small Petri dishes (diameter = 6 cm) with one honey bee larva for every five mites were placed back to the incubator under the same temperature and humidity conditions (34 °C and 65% of relative humidity). Mite mortality was determined after 24 and 48 h after incubation with a honeybee larva (as nourishment) [46]. Five experimental replicates were established for each exposure time; in total 140 mites were tested.

### 2.5. Repellent Effect of the EO towards the Mites

An amount of 0.3 g of essential oil was mixed with 30 g of liquid wax (concentration 1%). The solidified wax was placed at one end of a hollow tube. At the other end, pure wax without essential oil was placed. A mite was then inserted through a hole in the tube from its apex and center [52].

The location of the mite was observed and recorded at 10-min intervals for a total of 90 min. According to Kraus et al. (1994), the most pronounced orientation effects can be observed during this period [53].

To evaluate the repellency of the tested essential oil, 20 experimental replicates were analyzed. For each assay, one *V. destructor* mite was used and placed in an arena, as explained in the materials and methods section. The subject’s behavior was observed for 90 min following his introduction.

### 2.6. Honey Bee Workers: Toxicity Evaluation

To determine the toxicity of EO for adult honey bees, a pool of random individuals was collected. To obtain a random sample of honey bees of different ages, the subjects that make up the pool came from different hive frames. In particular, after being stirred in a container, the bees were sprayed with water to prevent flight and mixed [54].

The randomly collected bees were processed for toxicity tests.

Five technical replicates were analyzed. For each trial, as suggested by Bava et al. (2021) [26], two 50 mL Falcon tubes were filled with 1.6 mL of acetone and essential oil. The amount of liquid to be inserted was determined in relation to the volume used in the toxicity tests for the mites. As for the viability tests of *V. destructor*, the tubes were rolled several times on the walls, to coat the walls with liquid and to evaporate the acetone contained in the solution. Once dried, five honey bees were transferred to the tubes that previously contained the essential oil solution [26].

Therefore, the Falcon tubes with honey bees inside were transferred to an incubator (34 °C and 65% of relative humidity). One hour after exposure, the honey bees were placed in cages (cylindrical plastic box, Ø = 90 mm, height = 100 mm), according to William et al. (2015) [54]. These cages were equipped with feeders (50% sucrose solution and water). The honey bees were then observed for the next 48 h. In addition, as in the fumigation tests for mites, twenty adult honey bees were exposed to essential oil vapors. A two-story cage (Ø = 90 mm, height = 100 mm) equipped with feeders (50% sucrose solution and water), was composed. In the lower part, a gauze soaked with essential oil was inserted, while, in the upper part the honey bees were housed [55].

The cages where placed in an incubator (34 °C and 65% of relative humidity) and the subjects were observed for 48 h. In both cases, abnormal behavior and/or mortality was detected.

### 2.7. Statistical Analysis

The graphical representations of the datasets were created with jmpSAS software (JMP^®^ Pro, 2014. Version 14, Cary, NC, USA) using the graph builder module. Kruskal-Wallis tests were performed using the statistical module of jmpSAS to assess the effect of the treatments. Dunn’s Multiple Comparison with Bonferroni correction was used to evaluate the significance of the difference between the different experimental groups.

## 3. Results

### 3.1. Phytochemical Profile

The essential oil from the fresh aerial parts of *O. heracleoticum* was extracted by steam distillation for 2 h with a Clevenger-type apparatus obtaining an extraction yield of 0.8% *w*/*w*. The essential oil obtained was analyzed for its chemical composition providing the results summarized in Table 1. Overall, 37 compounds were identified. 15 components belong to the group of monoterpenic hydrocarbons, of which α-ocimene (5.7%) and sabinene (3.8%) are the most abundant chemical constituents. Furthermore, β-ocimene and mircene were detected at rates greater than 2%, and 4-carene and thujene at rates greater than 1%.

Nine oxygenated monoterpenes were also identified, including linalyl acetate (17.6%) and linalool (10.0%) which represent the most abundant essential oil components, followed by linalool propionate (6.6%) and eucalyptol (3.7%). Other oxygenated monoterpenes have been identified with percentages ranging from 0.2% to 0.7%. Moreover, 13 sesquiterpenes were also detected, of which bicyclosesquiphellandrene and β-bisabolene are the most abundant compounds (3.6% and 1.8%, respectively).

### 3.2. Acute Toxicity

The average neutralization percentages achieved after exposure to the different concentrations of essential oils and the relative standard deviations are reported in Table 2. In addition, the result graphical representation (Figure 1) shows how many mites, out of a total of 5 that made up each replicate (*n* = 10), were neutralized on average, at each concentration of EO when compared to the negative and positive control.

As reported in Table 2, the percentages of inhibition ranged from 54% to 91% depending on the concentration of EO used. Comparing the effect of each concentration with the negative control (acetone) the calculated *p*-values were always below 0.01 (Dunn’s Multiple Comparison with Bonferroni correction). The percentages of inhibition were very similar to the inhibition measured with the same concentrations of Amitraz.

### 3.3. Fumigant Toxicity

As can be seen, the toxicity of EO enhanced drastically in relation to the time of exposure (Figure 2). Figure 3 and Table 3 report the percentage of mortality reached at 24 and 48 h for the different exposure times. After 15 min of exposure to essential oil vapors, an average mortality rate of 8% at 24 h was registered, which increased to 12% at 48 h. A 30-min exposure resulted in a mortality rate of 28% at 24 h which reached 32% at 48 h. After one day, a 45-min exposure, resulted in a mortality rate of 56% of the subjects tested which rose to 72% after two days. Finally, the mortality rate at the maximum exposure time (90 min) was 60% and 84% at 24 and 48 h, respectively. The comparison of mortality percentages after 48 h between the exposure of 15 min and the exposure of 45 and 90 min was statistically significant (*p* = 0.0116 and *p* = 0.0015, respectively, calculated with Dunn’s Multiple Comparison with Bonferroni correction). Fumigation tests support the findings of Sabahi et al. (2017) [45]. Indeed, a longer exposure time of mites to the vapors of essential oil allows a better control of the infestation rate, resulting in a mortality related to subacute toxicity phenomena.

### 3.4. Repellent Activity

In general, it has been observed that the behavior of the mite is not influenced by the essential oil. The mites move indifferently in the direction of the object treated with the oil and in the opposite direction. The results are summarized in Table 4. No statistically significant differences were observed between the direction choices.

## 4. Discussion

The widespread phenomena of drug resistance and the strict legislation on chemical residues in food of animal origin require a remodeling of pharmacological treatments in farms. In this regard, medical treatments based on natural substances can represent a valid, safe and effective alternative therapeutic approach.

Essential oils isolated from many plants contain several bioactive compounds with different properties that can influence the behavior and vitality of insects [56,57]. These compounds have been shown regulatory or inhibitory effects on growth, development, reproduction and orientation of insects. These actions are often associated with other admirable properties. This aspect can extend the spectrum of action of essential oils to other pathologies in addition to parasite control as has been investigated in this study [40]. Some essential oils have been shown to be effective against American Foulbrood and the chalkbrood disease [29,58]. Therefore, the use of essential oils could be beneficial in several ways. These natural compounds can be proposed for the development of safe, effective and fully biodegradable insecticides.

In this study, the beneficial properties of the essential oil of *Origanum heracleoticum* were investigated. The present research study does not constitute a simple scientific pretension but is extremely important in the light of the knowledge acquired over the years. Particularly relevant is the excellent acaricidal efficacy of the essential oil of oregano which, in all the concentrations tested, was greater than that of the positive control. In the contact tests, the toxic compounds must penetrate the cuticular barrier to act, while, in case of fumigation they must be inhaled by the arthropod/insects. In our experiments, the essential oil of *O. heracleoticum* was effective both by contact and by fumigation. The results of our study are comparable to those of Hybl et al. (2021) [59] which verified the efficacy of the essential oil of *Origanum vulgare* against adult female *Varroa* mites, in laboratory conditions. Indeed, in their in vitro study, Hybl et al. (2021) used the same test employed in our study (glass-vial residual bioassay) to evaluate the acaricidal activity.

Importantly, Hybl et al. (2021) [59] estimated the mortality of *Varroa* mite after 2 and 4 h of essential oil exposure, with a 100% mortality rate. In the present study, following the indications of Bava et al. (2021) [26], the mortality rate was evaluated one hour after EO exposure, reaching a mortality percentage of 90.9% (2 mg/mL concentration). The choice of time is particularly important, as mites are sensitive to artificial environments. In fact, mites suffer more from hunger and water loss if kept away for a few hours from their natural habitat [60].

Previous studies on other *Origanum* species have shown how the efficacy of a pharmacological treatment is influenced by the method of administration and the duration of exposure to the treatment. The choice of the most effective pharmacological method of administration of the drug cannot ignore the knowledge of the methods of action involved. In this regard, important scientific evidence is reported in the studies conducted by Sabahi et al. (2017) about exposure times [45]. The authors applied oregano oil (Sigma^®^, Missasauga, ON, Canada) to hives using electric vaporizers (in vivo study), obtaining a reduction in the degree of infestation close to 97.4% [19]. These results are markedly better than those obtained by Romo-Chacón et al. (2016) with an average mortality between 57–74% by soaking a cotton towel with a solution containing oregano (*Lippia berlandieri*) essential oil [61]. A high mortality rate was obtained in tests performed by Gal et al. (1992) [62] which, in any case, resulted lower than that obtained by Sabahi’s group [35]. Therefore, the constant production of essential oil through fumigation for two weeks results in a more efficient treatment. In our study, fumigation tests gave similar results, confirming how longer exposure times resulted in higher mortality rates 48 h after application. However, in field conditions, various factors can influence the overall effectiveness of the treatment. These include the presence or absence of brood and the external environmental temperature [26]. Furthermore, the composition in active ingredients of the aromatic plants slightly varies depending on the time of harvest, the conditions of cultivation and how the plant is collected and stored [63]. The extraction method can also affect the final chemical profile of the essential oil extracted from a specific botanical species. We believe that the characterization of the phytochemical profile of the tested substances is a fundamental element that must always be associated with toxicity studies. Comparative studies of the phytochemical profiles and toxic activity of an essential oil derived from a particular botanical species, grown under different conditions, could help to better understand the effectiveness of the essential oil.

For this reason, in our study the chemical characterization of the essential oil tested was performed. The study of the phytochemical profile is particularly important as the species tested in our study (*O. heracleoticum*) is different from those tested in previous in vitro and in vivo tests. In our opinion, much of the scientific value of this work lies in this element. As can be seen in the characterization table, the most represented monoterpene molecules in the essential oil tested are linalyl acetate, linalool propionate and eucalyptol. The monoterpenoids are volatile compounds to which the fumigant activity is certainly referred. The fumigant action of essential oils has been investigated in several studies aimed at controlling parasites of stored products, lice, ticks and mites such as *Psoroptes* and *Sarcoptes scabiei* [64,65,66]. These studies have led to different products currently on the market based on these monoterpenoids.

Our results demonstrate that oregano oil can achieve a significant acaricidal effect against *V. destructor* in vitro. The assessment of acute toxicity by direct contact of the mite with a treated surface has been successful in terms of efficacy. As can be easily understood, the higher concentrations were more toxic to the mites and resulted in greater mortality phenomena. To our knowledge, the system used in the present study to perform the in vitro fumigation tests has never been used before. We found that, maintaining the concentration of the oil used constant (1 mg/mL), the mortality of the mites progressively increased as the time of exposure to fumigant vapors increased. Instead, the repellency tests revealed results with more difficult interpretation. Overall, there was no repellent effect of the presence of EO in wax. *V. destructor* moves indifferently in the direction of the material treated with the essential oil and in the untreated one. However, this lack of repellent ability may not represent an adverse effect of oregano oil. Indeed, the low repellency can favor the opportunities of contact of the parasite with a possible pharmacological formulation that exploits the contact toxicity of the active ingredients as well as the toxicity of evaporation.

## 5. Conclusions

In recent years, the phenomena of drug resistance are becoming more and more widespread. Essential oils can be a valid alternative to synthetic acaricides for the control of varroatosis in beekeeping [21,22,24,67,68]. Essential oils are environmentally friendly and, as their mechanisms of action are involved in different molecular pathways, it is rare for treated parasitic populations to develop resistance mechanisms. In particular, the essential oil tested has provided data in line with those reported in other works previously published for other species of oregano [45,61,62,69].

The use of *Origanum heracleoticum* essential oil for the varroatosis control could represent a valid alternative to the use of synthetic drugs. Therefore, it is important to investigate in field tests the most effective methods of administration, which can ensure a lasting persistence of the active ingredient, with a gradual release over time. In conclusion, this study is part of the green veterinary pharmacology [70], a branch of the pharmaceutical veterinary that, nowadays, must be implemented to reduce the phenomena of drug resistance and the persistence of residues in the environment. The advantages on animal welfare and health are indisputable, also linked to the consequent reduction of the transmission of bacteria and viruses, such as deformed wing virus (DWV) and slow bee paralysis virus (SBPV) by *V. destructor* mite to honey bees [71,72,73,74,75].

## Figures and Tables

**Figure 1 vetsci-09-00124-f001:**
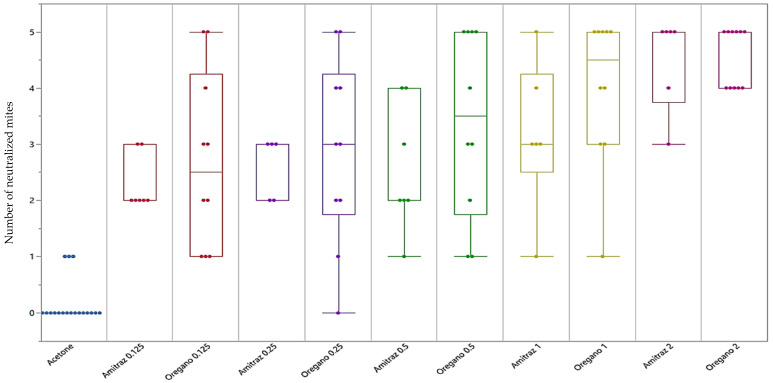
Representation of the effect of Oregano (*O. heracleoticum*) essential oil tested on mites, at a concentration of 0.125 mg/mL (*p* = 0.0058), 0.25 mg/mL (*p* = 0.0019), 0.5 mg/mL (*p* ≤ 0.001), 1 mg/mL (*p* ≤ 0.001), and 2 mg/mL (*p* ≤ 0.001), compared to the negative control (Acetone). The comparison of the negative control with the positive control retrieved the following results (Amitraz; 0.125 mg/mL, *p* = 0.1665; 0.25 mg/mL; *p* = 0.140; 0.5 mg/mL, *p* = 0.052, 1 mg/mL, *p* = 0.0068 and 2 mg/mL, *p* ≤ 0.001). *p*-values were calculated with Dunn’s Multiple Comparison with Bonferroni correction.

**Figure 2 vetsci-09-00124-f002:**
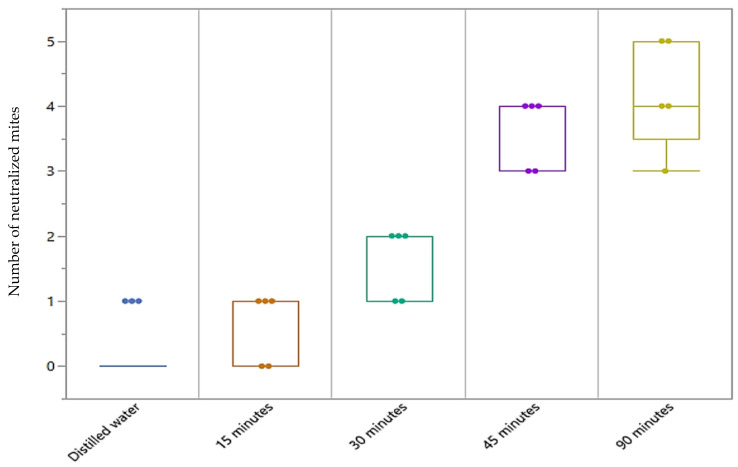
Representation of mites recorded as neutralized after 48 h of exposure at each time interval (15-min exposure vs. 30 min exposure, *p* = 0.98; 15 min exposure vs. 45 min exposure, *p* = 0.0116; 15-min exposure vs. 90 min exposure, *p* = 0.0015, respectively, calculated with Dunn’s Multiple Comparison with Bonferroni correction).

**Figure 3 vetsci-09-00124-f003:**
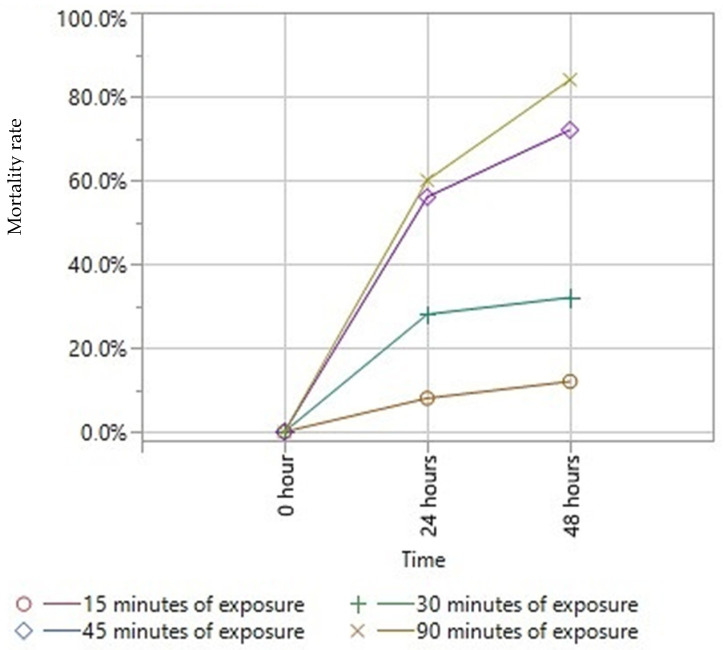
Mortality rates obtained with fumigation tests at 24 and 48 h.

**Table 1 vetsci-09-00124-t001:** Phytochemical profile of *O. heracleoticum* essential oil.

Compound	Rt ^(a)^	RAP ^(b)^
*Monoterpene hydrocarbons*		
Thujene	6.337	1.2
α-Pinene	6.580	0.8
Sabinene	7.483	3.8
β-pinene	7.557	0.6
Myrcene	7.803	2.0
α-Phellandrene	8.117	0.2
3-Carene	8.215	Tr ^(c)^
Terpinolene	8.300	Tr
4-Carene	8.449	1.4
β-Ocimene	8.735	2.2
Limonene	8.803	0.2
α-Ocimene	8.883	5.7
γ-terpinene	9.089	0.4
α-Fenchene	9.558	0.4
1,5,8-*p*-Menthatriene	10.209	0.2
*Oxygenated monoterpenes*		
Linalool	9.752	10.0
Eucalyptol	8.632	3.7
4-Thujanol	9.300	0.2
Terpinyl acetate	10.918	0.4
Linalool propionate	11.107	6.6
Methylthymol	11.621	0.5
Linalyl Acetate	12.010	17.6
Neryl acetate	13.078	0.4
Geranyl acetate	13.278	0.7
*Sesquiterpenes*		
Copaene	13.341	Tr
Bourbonene	13.427	0.7
β-Caryophyllene	13.678	Tr
β-Selinene	14.164	0.6
α-caryophyllene	14.187	0.7
Alloaromadendrene	14.239	0.2
γ-cadinene	14.353	Tr
Calarene	14.410	Tr
β-Cubebene	14.439	Tr
Bicyclosesquiphellandrene	14.456	3.6
β-Bisabolene	14.627	1.8
Farnesene	14.656	Tr
δ-Cadinene	14.810	0.3

^(a)^ Retention time (as min). ^(b)^ Relative area percentage (peak area relative to total peak area in total ion current (TIC)%). ^(c)^ Traces percentages < 0.1%.

**Table 2 vetsci-09-00124-t002:** *V. destructor* neutralized percentage and SD (±) after the treatment with *O. heracleoticum* essential oil, acetone and Amitraz.

Concentration (mg/mL)	Oregano *(O. heracleoticum)*	Acetone *(− control)*	Amitraz *(+ control)*
0.125	54 ± 31.3	3.16 ± 7.5	45.7 ± 9.8
0.25	58 ± 33.3		52 ± 11
0.5	68 ± 32.9		51.4 ± 22.7
1	80 ± 26.7		63.3 ± 26.6
2	90.9 ± 10.4		90 ± 16.7

**Table 3 vetsci-09-00124-t003:** Mean and SD (±) percent of *V. destructor* mites that were examined and found dead at 48 h, after each interval of exposure to oregano *O. heracleoticum* essential oil (EO) vapors.

Fumigation	15 min	30 min	45 min	90 min
*O. heracleoticum* EO	12 ± 11	32 ± 11	72 ± 11	84 ± 16.7

**Table 4 vetsci-09-00124-t004:** Percentage of times *V. destructor* mites were found in one tube location compared to another, after 90 min.

Oil	Wax + Oil	Pure Wax	Centre of the Tube	χ^2^
*O. heracleoticum* EO	35%	45%	20%	0.42 *p*: 0.5186

## Data Availability

Data available on request due to privacy restrictions.

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
