# Peer review of "Green Veterinary Pharmacology for Honey Bee Welfare and Health: Origanum heracleoticum L. (Lamiaceae) Essential Oil for the Control of the Apis mellifera Varroatosis"

_vetsci, 2022, doi:10.3390/vetsci9030124_

Round 1

Reviewer 1 Report

The manuscript was excellently written and is a very important contribution to the field of honey bee health and mite control. The experiments were well designed and conducted, BUT I must recommend that the manuscript be rejected and sent back to the authors to perform statistical analysis of their data. Modern scientific investigations based upon quantitative data as in this paper must be tested using statistical methods. It is only when this is done that valid conclusions can be made. I did not make any comments on the manuscript after the Methods Section.

Reviewer 2 Report

This study aims to test toxicity by direct and fumigation application of origan essential oil to the bee parasite Varroa destructor, and its host Apis mellifera. The authors associated lab experiments with proper analysis of the oil components. Overall the study is well designed and the problematic interesting, but they are several major points to improve before any publication I think (detailed bellow). English should be checked by a proper native speaker as sentences often lack fluidity.

-references : there are a LOT of assumptions you are making in your manuscript that lack proper references ; listed thereafter

      -l61 after parasite

      -l65 after beekeepers

       -l67 after oils, plus give axamples of other essential oils used

        -l77 after colony

       -l90 after larvae

       -l97 after plant

      -l100 after phenomena

      -l333 after vitality

-please check the whole document for italic for species names !

-please place the paragraph starting with "it"- until "known[36]" at line 111 between "species." and "there are", and replace the "it" with what it means (oregano ? or a special species ? )

-l130 you talk about another experiment from Sabahi et al, could you precise which species of origano was used in their study ?

-you talk about a lot of different origano, and in the Sabahi et al study you cite l128, could you precise if it ws the same origano species that you use in your study ? if not which one ?

_l149-150 please give more details on rearing conditions for your mites (Temperature, humidity, luminosity, petri dishes size, bee larvae collected when, renewed how often, feeded how ? 

-l155 please provide GPS data for sampling sites

-l221 precise the incubator temperature and humidity conditions 

l246 precise the cage dimensions

-l247 say "the honeybees were exposed to essential oil vapours in a two story cage equiped with feeders." how many honeybees exposed in the same time, feeded with what, which temperature and humidity conditions ?

-very important point : you need to add a "statistical analysis" section in you material and methods section, in which you will list the tests and models used to get your results. (and put the l279 in it )

-figure 1 legend : why didnt you put all the box plots a and b in the same graph ? is there a scientific reason ?

-l293 you say "it was agreed to use the concentration of 1mg", but why ? because of preliminary results ? a reference ? the use in the field ? what is the reason of this choice ?

-did you check/standardized  for  varroa stages/sex in your tests ?

-l294 how did you choose the incubation time ? is it in relation with litterature or field use ?

-figure 2 please be coherent with figure 1 and represent your results in boxplots too.

-fig3 put the scale bars, and change 0 "minute" in 0 "hour"

-l371 you say that the composition in active ingredients can vary with a lot of parameters, and so that is why you chemically looked at the oil composotion. ok now do you think that you can compare your results with theirs ? You could discuss the importance of characterize oils at different conditions to improve the understanding of the oil efficiency.

-i never saw this before : you litterally copy pasted the descriptive text of what was supposed to be in the aknowledgements section without filling it yourself... 

minor points :

-add dates after names of scientific cited in the text

-remove "the month of" before September l139

-remove the "," after "in order"

-

Round 2

Reviewer 1 Report

The authors did attempt to conduct statistical analysis, but used pairwise t-tests to evaluate differences from there negative control. This analysis, while not being absolutely wrong is not appropriate for the experimental designs that the authors have conducted. The authors need to assess overall treatment main effects and interactions between dose and compound where appropriate

The experimental designs should be analyzed with Analysis of Variance. The reviewer recommends that the authors consult with a statistician and correctly analyze their data.

Reviewer 2 Report

Thank you for your revised version of this manuscript, the quality of the manuscript has improved substancially, both in the text and the figures ; but it is not quite yet ready for publication. Please find bellow other revisions.

l30-32"The aim of this study was to assess the contact toxicity, the fumigation efficacy and the repellent effect of Origanum heracleoticum L. essential oil (EO) " please add "against Varroa destructor mites" afterwards

l69 please change "amitraz" to "Amitraz" 

l298 please give the statistical test used in brackets

l265-268 I appreciate that you add the statistical analysis section, but could you be more precise and specify which statistical tests you used when (student test? Chi test?) did you test for normality of you data ? 

figure 1 is better, thank you, but you should improve it further a bit by using colours more logically (colors in graphs have to have a meaning, it's not just esthetic): please use the same colors for each group modality to ease the reading of the figure. Moreover you say "comparison" in the legend, but you therefore have to specify which statistical comparison test you used, and add the statistical category letters upper each box to illustrate for significative differences.

l316-320 "The fumigation toxicity tests were conducted by exposing 5 mites to the vapors produced by 40µl of 1 mg/mL essential oil (water dilution) into a final volume of 2 ml (final concentration 20mg per litre of air). Four exposure times were evaluated: 15, 30, 45 and 90 minutes[46]. After each exposure, mite mortality was recorded at 24 and 48 hours after incubation with a honeybee larva (as nourishment). A total of 5 experimental replicates were analyzed for each exposure time. In total 140 mites were tested" all this should be in the material and method section, not in the results

L 321-323 in your description of the result please give more pertinent information, you should not write "Figure 2, clearly shows what happened to the mites over time." say for example instead "the accute toxicity of EO enhanced drastically with the time of exposure (Figure 2)" the same goes for figure 3 : you must interprete your graphs.

Please use the same colors codes in figure 2 and figure 3, and homogeneise the figure styles (figure 3 all writtings are in capital). in Figure 2 add "mites" to "neutralised" in y axis

l336 please precise the statistical test you use before the pvalue (student test ? Chi2 ?)

I am sorry but I dont see the point of the table 3, a one line table is really not usefull... I would suggest to integrate the values in the text instead, in a sentence.

l342-345 "To evaluate the repellency of the tested essential oil, 20 experimental replicates were analyzed. For each assay, one V. destructor mite was used and placed in an arena, as explained in the materials and methods section. The subject's behavior was observed for 90 minutes following his introduction" idem, this should not be in the results section, but in the material and methods one

the same than for table 3, I would suggest for you to integrate the table 4 in the text, as a one line table makes not much sense.

l386-390 wrong lettering (calibri style)

l403 "These include the presence or absence of brood and the external environmental temperature " please add some reference here to illustrate your point

 l432 "Overall, the behavior of the mites does not appear to be affected by the essential oil of oregano". I would rather suggest to write that there was no repulsive effect of the presence of EO in wax (because written like this this is confusing, it seems to be in contradiction with the mortality observed ). or You could also just remove this sentence as you already give more details just after it.

there lacks an "aknowledgements" section to have more details on the fundings linked to this paper

Round 3

Reviewer 1 Report

The authors have improved the manus rips by applying appropriate statistical analysis to the data. I have attached a copy of the manuscript that has minor suggestions on the grammar. I now feel that the paper is ready to publish after a few grammatical errors are corrected.

Reviewer 2 Report

Thank you for taking in account the last corrections, I'am happy with the manuscript to be accepted for publication now, please find bellow a few minor corrections.

l182 line feed missing after the section title

figure 1 and figure 2 please add "number of " before "mites neutralized" on the y axis

figure 3 y axis title missing. legend : "Percentage representation of Mortality rates obtained with fumigation tests at 24 and 48 hours".

litterature

ref 12, 13, 32, 36, 37, 38 , 39, 46, 49,  year is not bold

"Maggi, M.D.; Ruffinengo, S.R.; Negri, P.; Eguaras, M.J. Resistance phenomena to amitraz from populations of the ectoparasitic mite Varroa destructor of Argentina., doi:10.1007/s00436-010-1986-8." please complete the ref (Parasitology Research volume 107pages1189–1192 (2010))
